# 0.6-V 1.65-µW Second-Order $G_m$-C Bandpass Filter for Multi-Frequency Bioimpedance Analysis Based on a Bootstrapped Bulk-Driven Voltage Buffer

Juan M. Carrillo [1,*] and Carlos A. de la Cruz-Blas [2]

1 Department of Electrical, Electronic and Automation Engineering, University of Extremadura, Avenida de Elvas s/n, 06006 Badajoz, Spain
2 Institute of Smart Cities, IEEC Department, Public University of Navarre, 31006 Pamplona, Spain
* Correspondence: jmcarcal@unex.es

**Abstract:** A bootstrapping technique used to increase the intrinsic voltage gain of a bulk-driven MOS transistor is described in this paper. The proposed circuit incorporates a capacitor and a cutoff transistor to be connected to the gate terminal of a bulk-driven MOS device, thus achieving a quasi-floating-gate structure. As a result, the contribution of the gate transconductance is cancelled out and the voltage gain of the device is correspondingly increased. The technique allows for implementing a voltage follower with a voltage gain much closer to unity as compared to the conventional bulk-driven case. This voltage buffer, along with a pseudo-resistor, is used to design a linearized transconductor. The proposed transconductance cell includes an economic continuous tuning mechanism that permits programming the effective transconductance in a range sufficiently wide to counteract the typical variations that process parameters suffer during fabrication. The transconductor has been used to implement a second-order $G_m$-C bandpass filter with a relatively high selectivity factor, suited for multi-frequency bioimpedance analysis in a very low-voltage environment. All the circuits have been designed in 180 nm CMOS technology to operate with a 0.6-V single-supply voltage. Simulated results show that the proposed technique allows for increasing the linearity and reducing the input-referred noise of the bootstrapped bulk-driven MOS transistor, which results in an improvement of the overall performance of the transconductor. The center frequency of the bandpass filter designed can be programmed in the frequency range from 6.5 kHz to 37.5 kHz with a power consumption ranging between 1.34 µW and 2.19 µW. The circuit presents an in-band integrated noise of 190.5 µV$_{rms}$ and is able to process signals of 110 mV$_{pp}$ with a THD below −40 dB, thus leading to a dynamic range of 47.4 dB.

**Keywords:** bandpass filter; bootstrapping; bulk-driven; linearized transconductor; quasi-floating gate; voltage follower

## 1. Introduction

The electrical bioimpedance technique allows for characterizing indirectly the properties of a biological media in a noninvasive way [1]. An AC excitation signal is applied to the impedance under test, $Z_{BIO}$, and the corresponding response is acquired by means of an instrumentation amplifier [2], conditioned and processed. This technique is being widely used nowadays to assist in the diagnosis of different diseases extended among the population as well as for monitoring physiological variables [3,4]. Frequently, the response of the sample is required to be repeated at different frequencies in order to obtain a more complete information, which is known as bioimpedance spectroscopy. The typical frequency range, known as dispersion range, varies from several hundreds of Hz to a few MHz. The frequency analysis can be carried out sequentially, by modifying the frequency of the excitation signal. Nevertheless, when the bioimpedance of the media varies rapidly, a multi-frequency analysis is required in order to obtain all the responses at the same

time. In this case, as illustrated in Figure 1, different AC excitation signals are generated and simultaneously applied to the impedance, being subsequently separated with the help of bandpass filter (BPF) sections, being the $G_m$-$C$ a flexible and suitable approach for monolithic integration [5–15]. The resulting solution is susceptible of being incorporated in an Internet of Things (IoT) platform [16]. Nevertheless, different specifications must be met for this purpose, which can be especially stringent in terms of total power consumption when the overall application is intended to be incorporated into a wearable device.

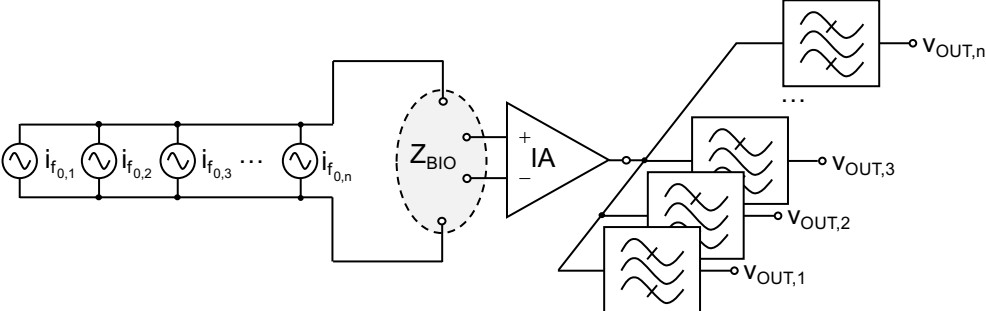

**Figure 1.** Block diagram of a multi-frequency bioimpedance system.

The bulk-driven technique is well-suited for low-voltage CMOS analog design, as it allows for operation with very low supply voltages and overcomes the non-zero threshold voltage constraint [10,17–25]. Indeed, in a bulk-driven transistor, the DC voltage required to switch the device on and the signal to be processed are decoupled and applied, respectively, to the gate and bulk terminal, which allows for providing and extending the input voltage range with respect to the conventional gate-driven device. Nevertheless, one of the main drawbacks of such technique is the reduction of the effective transconductance, due to the lower value of the bulk transconductance, $g_{mb}$, as compared to the gate transconductance, $g_m$. As a consequence, an increase of input-referenced magnitudes, such as the offset voltage or the noise, takes place. Different techniques have been proposed to electronically enhance the effective transconductance of a bulk-driven transistor, consequently increasing area and power consumption [26,27].

In this contribution, the application of a bootstrapping effect to a bulk-driven MOS transistor to increase its intrinsic voltage gain is proposed. The technique has been used to design a low-voltage voltage buffer, in which the noise contribution is reduced and the linearity is increased. The voltage buffer has been incorporated in the implementation of a linearized transconductor, which, in turn, is the basic building block of a second-order $G_m$-$C$ BPF aimed to signal separation in a multi-frequency bioimpedance measurement system. All the circuits have been designed in 180 nm CMOS technology to operate with a 0.6-V single supply. The rest of the manuscript has been organized as follows: In Section 2, the voltage buffer is described and analyzed, whereas simulated results are used to confirm its principle of operation. The design of the linearized transconductor is detailed in Section 3 and the implementation of the filter is presented in Section 4. Simulated results are provided in Section 5 and conclusions are drawn in Section 6.

## 2. Boostrapped Bulk-Driven Voltage Follower

### 2.1. Bulk Driven Buffer: Simulation and Analytical Results

Figure 2a illustrates a conventional bulk-driven flipped voltage follower, where the input voltage is applied to the bulk of transistor MD, a bias voltage $V_{BIAS}$ is applied to its gate, and the output voltage $V_{OUT}$ is obtained at the source. A negative feedback loop is established around transistors MF and MD, which forces the current $I_B$ via the constant voltage $V_{BN}$ to flow through the drain of device MD, and ensures a very low output resistance.

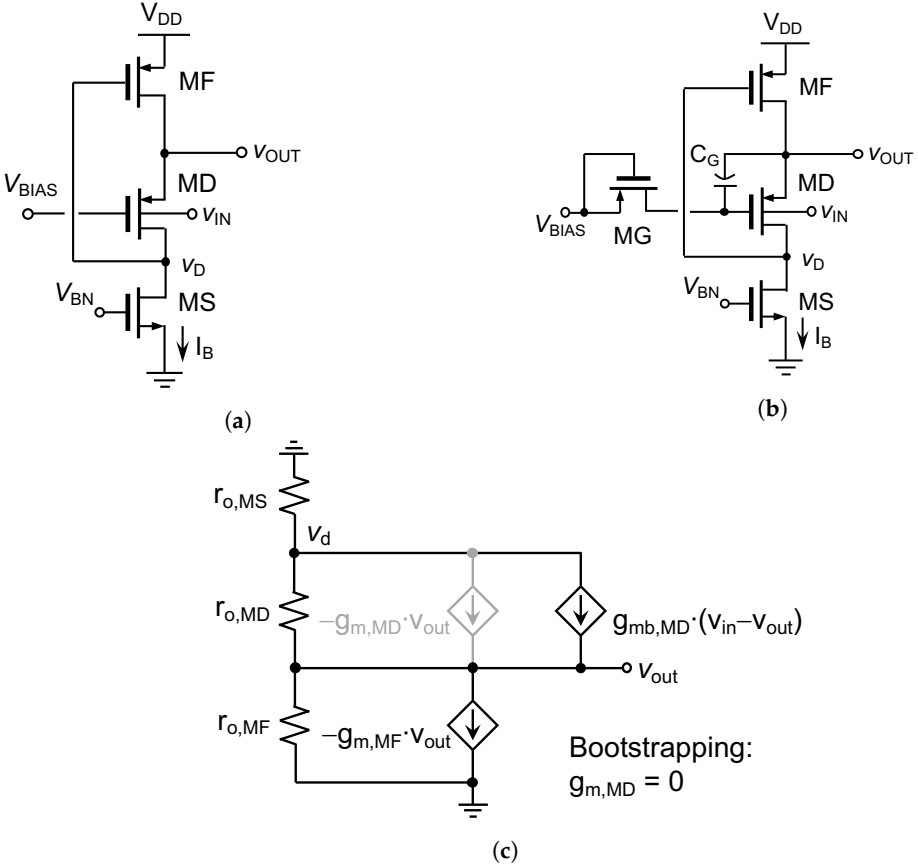

**Figure 2.** Bulk-driven FVF cell: (**a**) conventional approach; (**b**) proposed bootstrapped version; and (**c**) small–signal circuit ($g_{m,MD} = 0$ for the bootstrapped case).

The proposed circuit is implemented by adding a capacitor $C_G$ between the gate and source terminals of MD and a cutoff transistor MG acting as a pseudo-resistor between $V_{BIAS}$ and the gate of MD, as shown in Figure 2b, in a similar way as in the quasi-floating gate transistor technique [28]. It is worth noting that these elements are the ones usually employed to design a bootstrapping circuit [29,30], but they are used here to cancel out the gate transconductance of transistor MD, i.e., $g_{m,MD} = 0$, thus enhancing the voltage gain of the cell.

Figure 2c depicts the equivalent small-signal circuit of Figure 2a and the main parameters of the cell are summarized in the second column of Table 1, where $g_{m,Mi}$, $g_{mb,Mi}$, and $r_{o,Mi}$ are the gate transconductance, the bulk transconductance, and the output resistance of transistor M$i$, respectively. In addition, $R_{D,MD}$ and $R_{S,MD}$ are the equivalent resistances seen from the drain and source terminals of MD, also respectively. The small-signal equivalent circuit of the buffer in Figure 2b is very similar to the one illustrated in Figure 2c, but due to the bootstrapping effect $g_{m,MD} = 0$. As a result, the corresponding small-signal expressions are modified accordingly for the proposed approach, as shown in the third column of Table 1. Note that, for the case of the voltage gain, the proposed circuit avoids the signal attenuation inherent in the bulk-driven technique. In return, the values of $R_{out}$ and $R_{S,MD}$ are incremented due to the cancellation of $g_{m,MD}$. On the other hand, the open loop gain is the same for both circuits, i.e., $g_{mb,MD} \cdot r_{o,MD}$, whereas the loop gain can be expressed as $(g_{m,MD} + g_{mb,MD}) \cdot r_{o,MD}$ and $g_{mb,MD} \cdot r_{o,MD}$ for the conventional and the bootstrapped version, respectively [31].

**Table 1.** Small-signal parameter comparison of the conventional and bootstrapped buffers.

| | Conventional | Bootstrapped |
|---|---|---|
| Gain | $\dfrac{g_{mb,MD}}{g_{m,MD}+g_{mb,MD}}$ | $\approx 1$ |
| $R_{out}$ | $\dfrac{1}{g_{m,MF}\cdot(g_{m,MD}+g_{mb,MD})\cdot(r_{o,MD}\|r_{o,MS})}$ | $\dfrac{1}{g_{m,MF}\cdot g_{mb,MD}\cdot(r_{o,MD}\|r_{o,MS})}$ |
| $R_{D,MD}$ | $\dfrac{1}{g_{m,MF}}$ | $\dfrac{1}{g_{m,MF}}$ |
| $R_{S,MD}$ | $\dfrac{1}{g_{m,MD}+g_{mb,MD}}$ | $\dfrac{1}{g_{mb,MD}}$ |
| Open loop gain | $g_{mb,MD}\cdot r_{o,MD}$ | $g_{mb,MD}\cdot r_{o,MD}$ |
| Loop gain | $(g_{m,MD}+g_{mb,MD})\cdot r_{o,MD}$ | $g_{mb,MD}\cdot r_{o,MD}$ |

## 2.2. Analytical and Simulated Results

In this subsection, analytical expressions and simulation results of the conventional and proposed buffer are provided. The simulations have been obtained using a standard 180 nm CMOS technology with the following aspect ratios for the common transistors $W_{MD}/L_{MD} = 20$ μm/1 μm, $W_{MF}/L_{MF} = 1$ μm/1 μm, $I_B = 100$ nA, set by a simple current mirror with $W_{MS}/L_{MS} = 4$ μm/1 μm. For the bootstrapped implementation, $C_G = 0.25$ pF and transistor MG ($W_{MG}/L_{MG} = 240$ nm/340 nm) is connected as a pseudo-resistor, implemented by a thick oxide device to obtain a larger value of resistance when it is compared to standard transistors. As a consequence, a lower operating cutoff frequency can be achieved. The supply voltage was set equal to 0.6 V; both cells were loaded with an output capacitor of 50 fF, and $V_{BIAS}$ was fixed to 0.1 V.

*Gain, area, and power consumption:* Figure 3 shows a comparison of the AC small-signal response of the conventional and the bootstrapped buffers. The technique operates properly for frequencies higher than 3 Hz, obtaining a gain of 0.21 V/V (−13.4 dB) and 0.92 V/V (−0.7 dB) for the conventional and the proposed cell, respectively. For obtaining operation at lower frequencies, capacitor $C_G$ should be made larger or the configuration of the pseudo-resistor could be modified to increase its value. In the case of the high cutoff frequency, the value for the proposed cell is lower as compared to the conventional solution, since the output resistance of the proposed cell has been increased. The small overdamping observed in the magnitude response of the proposed circuit at frequencies slightly higher than 1 MHz can be easily cancelled by connecting a very small capacitor at the drain terminal of the driver transistors MD in Figure 2b. In any case, it does not affect the stability of the feedback loop implicit in the buffer. The power consumption is the same in both designs, 60 nW (not including the bias circuits), whereas in terms of silicon area, the proposed cell is twice as large as the conventional technique due to the presence of capacitor $C_G$. However, larger capacities (in the order of tenths of pF) will be used in the final application, thus making this increase in area not very significant. In addition, it is worth mentioning that, in the used technology, metal–insulator–metal capacitors can be placed on top of the active devices, which allows for reducing the total area occupation of the voltage buffer.

Figure 4 shows the voltage gain of the conventional and the bootstrapped buffers as a function of the input differential-mode (DM) voltage in a range from −200 mV to 200 mV with respect to a common-mode (CM) voltage of 300 mV. Note that the gain of the proposed cell is more than four times higher than that of the conventional cell in the voltage range between −150 mV and 150 mV, and it is much closer to unity. In addition, the proposed cell has a more constant response than the conventional cell, leading to a more linear behavior, as it will be demonstrated next.

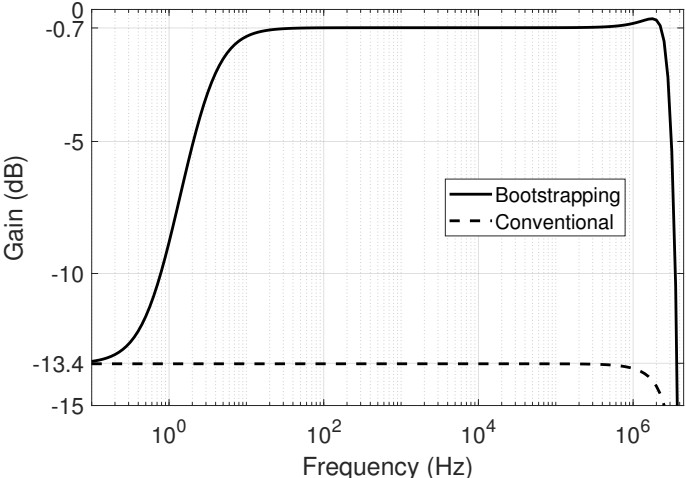

**Figure 3.** Frequency response comparison of the conventional and bootstrapped buffers.

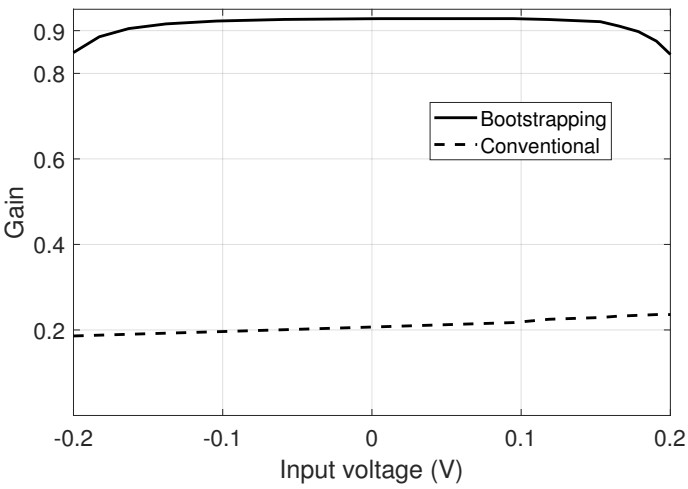

**Figure 4.** Gain versus input DM voltage of the two voltage buffers.

*THD analysis:* Considering that the PMOS transistors in Figure 2 operate saturated in the weak inversion region, and neglecting the channel length modulation effect, their drain current can be defined as [22]

$$i_D = I_T\left(\frac{W}{L}\right)exp\left(\frac{V_{SG} + V_{th}}{nV_T}\right)\left[1 - exp\left(\frac{V_{SD}}{V_T}\right)\right] \tag{1}$$

where $I_T$, $V_{th}$, $n$, and $V_T$ are the technology current, the threshold voltage, the subthreshold slope, and the thermal potential, respectively. In a bulk-driven transistor, the signal is implicit in the threshold voltage, which can be expressed as

$$V_{th} = V_{th0} - \gamma_P\left(\sqrt{2\phi + V_{BS}} - \sqrt{2\phi}\right) \tag{2}$$

where $V_{th0}$ is the threshold voltage when $V_{BS} = 0$ and $\phi$ and $\gamma_P$ are fabrication process constants. It is worth pointing out that, for a PMOS transistor, the values of $V_{th}$, $V_{th0}$, and $\gamma_P$ are negative. Using these expressions, it is possible to find a closed-form relationship between $v_{OUT}$ and $v_{IN}$ for the circuits in Figure 2. Indeed, the large-signal input/output voltage expression for the conventional bulk-driven FVF cell is the solution of a quadratic function that can be written as follows:

$$v_{OUT} = \frac{-(2A + \gamma_P^2) \pm \sqrt{\gamma_P^4 + \gamma_P^2(4A + 8\phi) + 4\gamma_P^2 v_{IN}}}{2} \tag{3}$$

with $A = -V_{BIAS} + V_{th0} + \gamma_P \sqrt{2\phi} - nV_T \ln\left(\frac{I_T}{I_S(W/L)}\right)$. An evident nonlinear behavior can be observed in the input/output transfer characteristic of the conventional voltage follower. On the other hand, the $v_{OUT} - v_{IN}$ transfer characteristic of the proposed buffer is inherently linear and given by:

$$v_{OUT} = 2\phi - \frac{A^2}{\gamma_P^2} + v_{IN} \tag{4}$$

As inferred from (4), the linearity of the proposed cell is improved since the AC signal at the source terminal of transistor MD is copied to its gate, allowing the input/output voltage relationship to become linear. As a consequence, the THD performance is better for the proposed bootstrapped buffer as compared to the conventional structure.

Figure 5 shows the simulated THD comparison for a sinusoidal input signal of 1 kHz with an amplitude swept from 10 mV to 250 mV. The dominant distortion contribution in both cases is due to the second-order harmonic. Note that the proposed cell has a THD lower than 1% ($-40$ dB) for input signals up to 180 mV, with a corresponding output voltage of 166 mV, whereas, for the conventional cell, an input signal of only 50 mV, corresponding to an output voltage of 10 mV, is allowed to achieve the same distortion level. This represents an increase of almost 5 and 20 times of the maximum input and output signal levels, respectively, that can be processed.

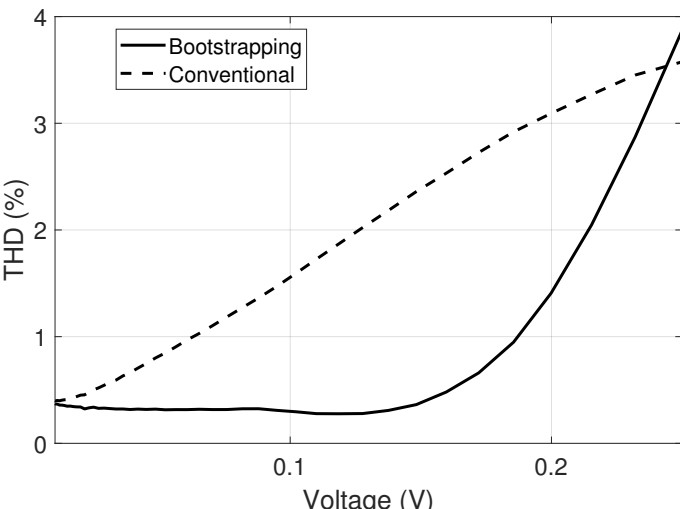

**Figure 5.** THD comparison.

*Noise response:* A straightforward analysis of the noise equivalent circuit of the conventional buffer reveals that the power spectral density of the input-referred noise is:

$$\frac{\overline{n_{iC}^2}}{\Delta f} = \frac{\overline{i_{n,MF}^2}}{\Delta f} \frac{1}{g_{m,MF}^2 g_{mb,MD}^2 (r_{o,MD} \parallel r_{o,MS})^2} + \frac{\overline{i_{n,MD}^2}}{\Delta f} \frac{1}{g_{mb,MD}^2} + \frac{\overline{i_{nb,MF}^2}}{\Delta f} \frac{(g_{m,MD} + g_{mb,MD})^2}{g_{m,MF}^2 g_{mb,MD}^2} \tag{5}$$

where the subscripts of the noise current sources are related to the names of the transistors in Figure 2. On the other hand, for the bootstrapped version of the voltage buffer, we have:

$$\frac{\overline{n_{iB}^2}}{\Delta f} = \frac{\overline{i_{n,MF}^2}}{\Delta f} \frac{1}{g_{m,MF}^2 g_{mb,MD}^2 (r_{o,MD} \parallel r_{o,MS})^2} + \frac{\overline{i_{n,MD}^2}}{\Delta f} \frac{1}{g_{mb,MD}^2} + \frac{\overline{i_{nb,MF}^2}}{\Delta f} \frac{1}{g_{m,MF}^2} \tag{6}$$

As it can be seen in (5) and (6), the first two noise contributions are equal because the ratio of $R_{out}$ to gain and $R_{S,MD}$ to gain are the same in both circuits. The difference relies on the last term, related to the ratio of $R_{D,MD}$ to gain, which is different in both implemen-

tations. Subtracting both equations and defining $g_{mb,MD} = \eta g_{m,MD}$ and $g_{mb,MD} = \lambda g_{m,MF}$, the extra noise for the conventional buffer is:

$$\frac{\overline{n_{iC}^2}}{\Delta f} - \frac{\overline{n_{iB}^2}}{\Delta f} = \frac{\overline{i_{nb,MF}^2}}{\Delta f} \cdot \frac{\frac{2\lambda^2}{\eta} + \frac{\lambda^2}{\eta^2}}{g_{mb,MD}^2} \tag{7}$$

In Figure 6, it is evidenced by simulations that the noise corresponding to the bootstrapped buffer is lower than in the case of the conventional solution, according also to the prediction in (7).

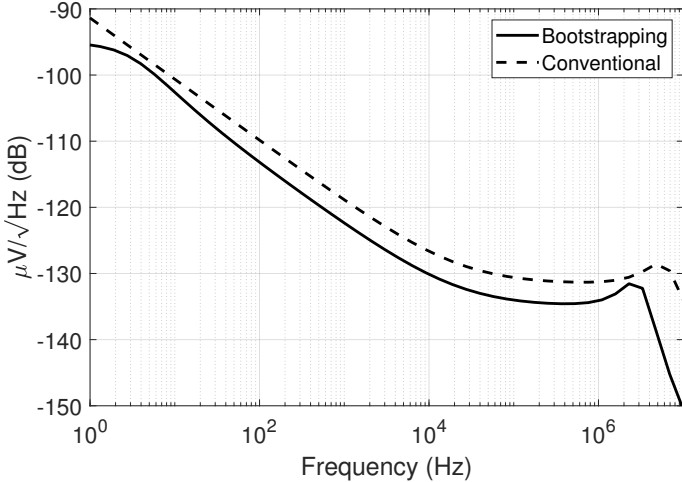

**Figure 6.** Noise comparison. The input power spectral density is represented in dB on the *y*-axis to illustrate more clearly the tendencies.

### 3. Proposed Linearized Transconductor

The circuit schematic of the proposed transconductor, consisting of a linearization resistor and two voltage followers, is illustrated in Figure 7. The input signals, $v_{IN}^+$ and $v_{IN}^-$, are applied to the bulk terminal of the driver transistors MD1 and MD2, producing a buffered replica of these voltages, $v_{IN,B}^+$ and $v_{IN,B}^-$, at their source terminal. The bootstrapping action applied to the bulk-driven transistors leads to a gain close to unity for the voltage followers, as detailed in the previous section. The corresponding DM signal, $v_{IN,B}^+ - v_{IN,B}^-$, is applied to a pseudo-resistor, implemented by transistors MR1 and MR2, where voltage-to-current (*V*-to-*I*) conversion takes place.

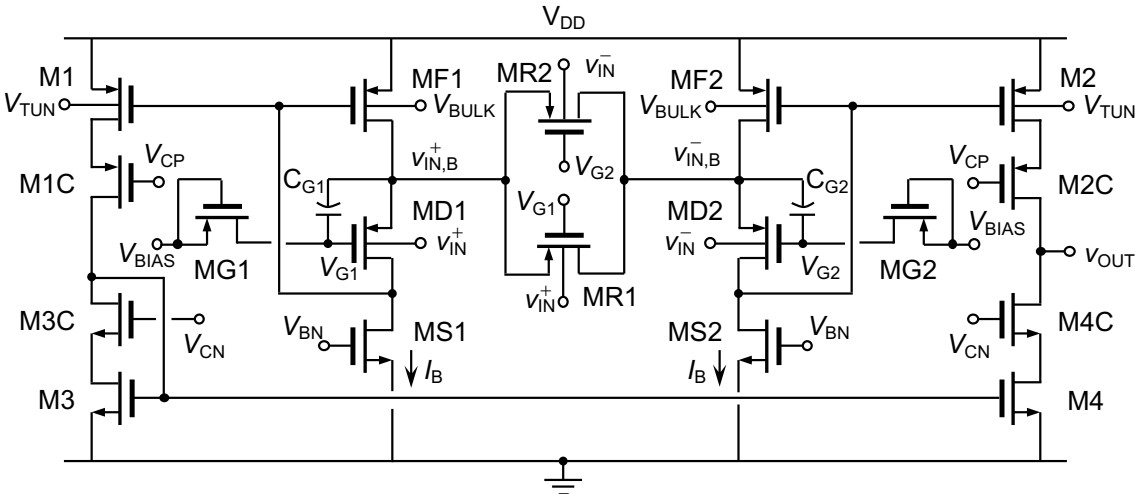

**Figure 7.** Proposed linearized transconductor.

Assuming that the parallel connection of transistors MR1 and MR2 leads to a resistor with an approximately constant value $R_{LIN}$ for small values of their source-to-drain voltage, the effective transconductance of the V-to-I converter has been determined by means of a hand analysis, and can be expressed as:

$$G_{m,eff} = \frac{2}{R_{LIN}} \cdot \alpha_{BD} \cdot \frac{1}{1 + \frac{2}{R_{LIN}} \cdot \frac{1}{g_{mb,MD}+g_{m,MD}} \cdot \frac{g_{o,MD}+g_{o,MS}}{g_{m,MF}}} \approx \frac{2}{R_{LIN}} \tag{8}$$

where $g_{mb,Mi}$, $g_{m,Mi}$, and $g_{o,Mi}$ are the bulk transconductance, gate transconductance, and output conductance of transistor M$i$, respectively, and $\alpha_{BD}$ is the intrinsic gain of the bulk-driven follower. In the case of a conventional bulk-driven FVF, $\alpha_{BD} = g_{mb,MD}/(g_{mb,MD} + g_{m,MD})$, causing a noticeable signal attenuation that leads to a transconductance degeneration. The signal attenuation can result adequate in a low-voltage environment, as it reduces the signal swing at the intermediate nodes of the transconductor. Nevertheless, this decrease of the effective input transconductance leads to an increase of input-referred magnitudes, such as the noise or the offset voltage. Alternatively, when the proposed bootstrapped bulk-driven FVF is used, it happens that $\alpha_{BD} \approx 1$ and, hence, there is an enhancement of the transconductance of the cell.

The response of the transconductor is linearized by connecting the bulk terminals of the transistors in the active resistor, MR1 and MR2, to the input terminals of the transconductor, $v_I^+$ and $v_I^-$, whereas the gate terminals are connected to the bootstrapping network in order to also benefit from this effect. This solution, first proposed in [32] and adapted to operate with bulk-driven transistors in [22], is modified here to also take advantage of the bootstrapping effect. Indeed, the common connection of the gate, source, and bulk terminals of transistors MD1-MR1 and MD2-MR2 in the core of the transconductor leads to equal $V_{SG}$ and $V_{SB}$ voltages for each pair of devices and, hence, to a linearized response that is also insensitive to variations in the input CM voltage [22]. The general expression of the drain current of a MOS transistor operated in the subthreshold region, given by (1), can be approximated by means of the Taylor series when the transistor operates in triode, i.e., when $v_{DS}$ is very small. In particular, the Taylor series can be truncated at the linear term, thus obtaining

$$i_{D,triode} = \frac{I_T}{V_T}\left(\frac{W}{L}\right)exp\left(\frac{V_{SG} + V_{th}}{nV_T}\right)v_{SD} \tag{9}$$

Similarly, the expression of the threshold voltage can be linearized as [23]

$$V_{th} = V_{th0} - (n-1)v_{BS} \tag{10}$$

Considering the expressions in (9) and (10), the output conductance of a MOS transistor biased in the subthreshold region and operated in triode can be written as:

$$g_o \equiv \frac{di_D}{dv_{DS}} \approx \frac{I_T}{V_T}\left(\frac{W}{L}\right)exp\left(\frac{V_{SG} + V_{th0} - (n-1)v_{BS}}{nV_T}\right) \tag{11}$$

As transistors MR1 and MR2 in Figure 7 are connected in parallel, the effective conductance of the composite structure, $g_{LIN} = R_{LIN}^{-1}$, is the sum of the individual conductances of both devices. Assuming that the signal $v_{BS}$ applied at the bulk terminals of devices MR1 and MR2 has a CM DC component, $V_{BS}$, and a purely DM signal contribution, $v_i$ and $-v_i$, respectively, the value of the linearization resistor can be approximated as:

$$R_{LIN} = \frac{1}{g_{LIN}} = \frac{1}{g_{o,MR1} + g_{o,MR2}} =$$

$$= \left[\frac{I_T}{V_T}\left(\frac{W}{L}\right)exp\left(\frac{V_{SG} + V_{th0} - (n-1)V_{BS}}{nV_T}\right) \cdot 2\left(1 + \left(\frac{(n-1)v_i}{nV_T}\right)^2 + \left(\frac{(n-1)v_i}{nV_T}\right)^4 + ...\right)\right]^{-1} \tag{12}$$

The odd-power terms of the signal cancel out each other, whereas the even-power terms are summed. Taking into account only the linear term of $v_i$ signal, the expression of the linearization resistor can be further approximated as

$$R_{LIN} = \left[ 2\frac{I_T}{V_T}\left(\frac{W}{L}\right)exp\left(\frac{V_{SG} + V_{th0} - (n-1)V_{BS}}{nV_T}\right)\right]^{-1}. \qquad (13)$$

The circuit section used to bias the transconductor is shown in Figure 8. In particular, voltages $V_{BN}$ and $V_{BP}$ are used to generate the different replicas of the biasing current $I_B$ required in the *V*-to-*I* converter. Furthermore, voltages $V_{CN}$ and $V_{CP}$ allow for biasing NMOS and PMOS cascode devices. An ultra-low-voltage environment connecting the gate of NMOS and PMOS cascode transistors to $V_{DD}$ and ground, respectively, seems to be a straightforward biasing solution leading to a reduction of the total current consumption. Nevertheless, appropriate bias conditions would be only ensured in typical mean conditions and at the nominal value of the supply voltage and the temperature. The use of the simple and well-known structure in Figure 8 allows for tracking PVT variations and translate them to the bias voltage of the cascode transistors. A similar situation arises in the biasing of the gates of the bulk-driven MOS transistors through the bootstrapping network, the reason why the DC signal $V_{BIAS}$ is also generated.

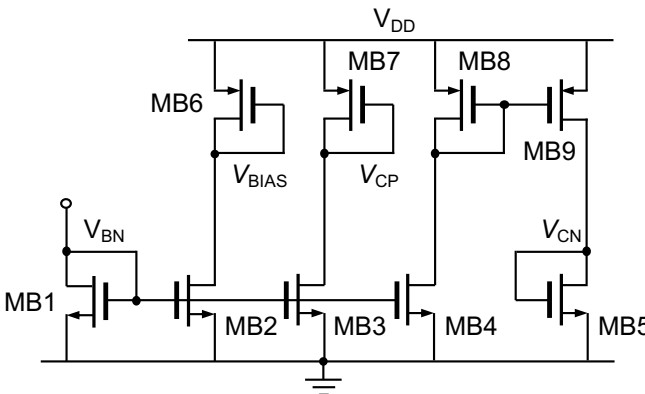

**Figure 8.** Circuit section used to generate biasing voltages and currents.

Conventionally, the transconductance of the *V*-to-*I* converter illustrated in Figure 7 is tuned by modifying the value of the tail current of the FVF cells. As current $I_B$ changes, the $V_{SG}$ of the driver transistors also does, modifying the effective value of $R_{LIN}$ and, hence, of $G_{m,eff}$. Here, a different tuning mechanism, based on controlling the gain of the PMOS current mirrors formed by transistors MF1-M1 and MF2-M2, is proposed. The bulk terminal of the input transistors of the current mirror, MF1 and MF2, is connected to a fixed DC voltage $V_{BULK}$, whereas a variable voltage $V_{TUN}$ is applied to the bulk terminal of the output transistors, M1 and M2. When $V_{TUN} > V_{BULK}$, the effective threshold voltage of the output transistors is higher and the current flowing though the output branch is lower, thus having a current attenuation. Conversely, for $V_{TUN} < V_{BULK}$, the effective value of $V_{th}$ of the output transistors of the current mirror becomes lower than that of the input transistors, obtaining a higher output current and, hence, a signal amplification. The voltage $V_{TUN}$ finds its upper bound in the supply voltage $V_{DD}$ and, theoretically, can be decreased until the ground level is reached. Nevertheless, considering that the source and the bulk of these transistors form a *pn* junction, deep forward biasing of this parasitic diode must be avoided. To this end, the exponential behavior of the current flowing through the bulk terminal of a PMOS transistor when the bulk voltage is changed has been considered in order to determine a practical lower bound for the tuning range of voltage $V_{TUN}$. In particular, in Figure 9, the bulk current of transistors M1 and M2 in Figure 7, $I_{BULK}$, is represented as a function of the tuning variable $V_{TUN}$. A current level equal to 1% of the biasing current, i.e., $0.01I_B$, has been selected as a reasonable limit in order to avoid deep forward operation

of the source-bulk *pn* junction of transistors M1 and M2. As a result, a value of 200 mV for $V_{TUN}$ is selected as the lower bound of the tuning variable.

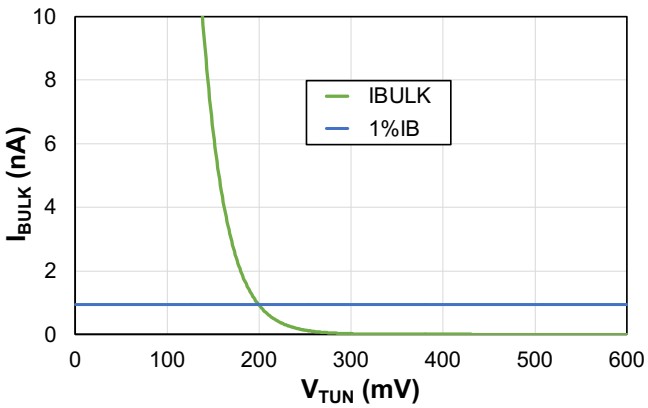

**Figure 9.** Bulk current over the tuning variable $V_{TUN}$.

## 4. Second-Order $G_m$-C Bandpass Filter

The second-order $G_m$-C BPF illustrated in Figure 10 has been implemented by using the linearized transconductor described in the previous section and depicted in Figure 7, which is based in turn on the bootstrapped bulk-driven voltage buffer shown in Figure 2b. The filter structure incorporates four transconductors in order to be able to set independently the center frequency, $\omega_0$, the gain at the center frequency, $|H(\omega_0)|$, and the quality factor, $Q$. In our application, only $\omega_0$ is intended to be swept, whereas $|H(\omega_0)|$ and $Q$ will have fixed values. Nevertheless, the configuration selected allows for keeping constant a given quality factor while the center frequency is swept. In addition, there is an additional degree of freedom in the structure that allows for maximizing the dynamic range of the BPF. Indeed, the other node in the filter, $v_{OUT,LP}$ in Figure 10, provides a lowpass response. The lowpass response presents an overdamping at the frequency of the poles that is a function of the quality factor selected for the BPF. As a consequence, a noticeable peak appears at that node at $\omega_0$, thus limiting the dynamic response of the overall biquad. This fact can be avoided with the structure illustrated in Figure 10, as the value of $Q$ can be set through the ratios of the active (transconductance) or the passive (capacitor) elements, which allows for decreasing the overall gain of the lowpass response, thus decreasing the maximum signal amplitude achieved at $v_{OUT,LP}$ at the center frequency of the BPF.

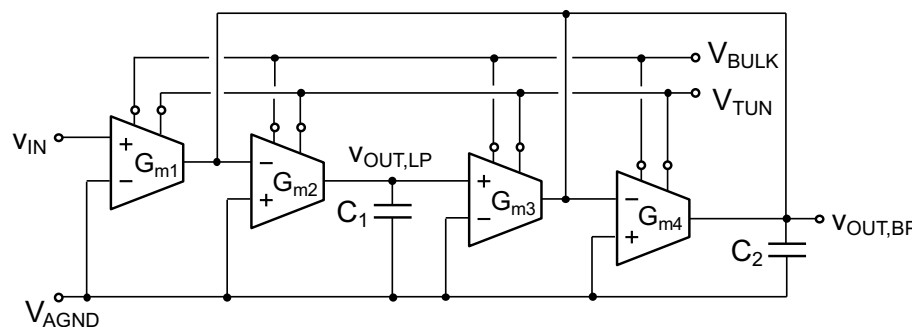

**Figure 10.** Second-order $G_m$-C bandpass filter.

The transfer function of the selected BPF can be written as:

$$H(s)_{BP} = \frac{\frac{G_{m1}}{C_2}s}{s^2 + \frac{G_{m4}}{C_2}s + \frac{G_{m2}G_{m3}}{C_1 C_2}} \tag{14}$$

where $G_{mi}$, with $i$ = 1 to 4, represents the effective transconductance of the $i$-th transconductor and $C_1$ and $C_2$ are integrated capacitors. The gain at the center frequency, $|H(\omega_0)|$, the center frequency, $\omega_0$, and the quality factor, $Q$, can be obtained from (14) in a straightforward manner and expressed as:

$$|H(\omega_0)| = \frac{G_{m1}}{G_{m4}} \tag{15a}$$

$$\omega_0 = \sqrt{\frac{G_{m2}G_{m3}}{C_1 C_2}} \tag{15b}$$

$$Q = \sqrt{\frac{C_2}{C_1} \cdot \frac{G_{m2}G_{m3}}{G_{m4}^2}} \tag{15c}$$

The intended application of the BPF is the separation of signals with different frequencies in a multi-frequency bioimpedance measurement system. Thus, the selectivity of the filter must be relatively high, which requires a moderately high value of the quality factor. A hand-analysis of the response at node $v_{OUT,LP}$ of the filter reveals that an optimal choice in order not to limit the dynamic range of the BPF response is obtained when $C_1 = C_2 = C$. Thus, the following equality has been established for the transconductances $G_{m2} = G_{m3} = k \cdot G_{m4} = k \cdot G_m$ so that the factor $Q$ is equal to parameter $k$. In addition, transconductors $G_{m1}$ and $G_{m4}$ have been sized to be equal, $G_{m1} = G_{m4} = G_m$, in order to have a gain at the center frequency equal to unity. Therefore, the expressions in (15a–15c) can be rewritten as:

$$|H(\omega_0)| = 1 \tag{16a}$$

$$\omega_0 = k \cdot \frac{G_m}{C} \tag{16b}$$

$$Q = k \tag{16c}$$

The factor $k$ has been achieved by properly sizing the pseudo-resistor in each transconductor, whereas the rest of the $V$-to-$I$ converter has been kept equal. The response of the BPF, in particular the center frequency, can be programmed by fixing voltage $V_{BULK}$ to an appropriate value and by tuning the value of the control voltage $V_{TUN}$ around it. For $V_{TUN} = V_{DD}$, the transconductors achieve their minimum transconductance value, thus leading to the lowest value of $\omega_0$. Conversely, when $V_{TUN}$ reaches the minimum reliable value, the $G_m$ is maximized and also is the value of the center frequency.

## 5. Simulated Results

The bootstrapped bulk-driven voltage buffer in Figure 2b, the linearized transconductor in Figure 7, and the second-order $G_m$-$C$ BPF in Figure 10 have been designed in 180 nm CMOS technology to operate with a single-supply of 0.6 V. The simulated results corresponding to the voltage buffer have already been provided in Section 2 in order to demonstrate its principle of operation and, hence, the metrics corresponding to the other two blocks are described here.

The sizes of the main transistors involved in the implementation of the linearized transconductor are reported in Table 2, whereas the value of capacitors $C_{G1}$ and $C_{G2}$ was set equal to 0.25 pF. The circuit was biased with a current $I_B$ = 100 nA and the value of the voltages $V_{BULK}$ and $V_{TUN}$ was nominally set equal to 400 mV. In addition, a load capacitor of 1 pF was connected to the output terminal. The transconductor was first characterized at low frequency, as the bootstrapped structure is not DC coupled. The effective transconductance, $G_{m,eff}$, was simulated and is represented in Figure 11 as a function of the input DM voltage when the value of the tuning variable $V_{TUN}$ is swept from 200 mV to 600 mV. As observed, the transconductance can be programmed in a range of approximately 5×, showing a linearized behavior, even though some dependence on the level of the input signal can also

be noticed, as predicted by (12). The open-loop frequency response of the transconductor is illustrated in Figure 12, where the magnitude and the phase of the voltage gain are represented. The low frequency corner due to the bootstrapping network is located at around 2.5 Hz, whereas the voltage gain in the low frequency band is 54.2 dB with a unity gain frequency is equal to 94.2 kHz and a phase margin of 85.6°. The low frequency corner achieved is compatible with the frequency range of interest in the intended application. If, for any reason, a lower cutoff frequency is required, a larger value for the gate capacitor $C_G$ or the pseudo-resistor MG in the bootstrapping network has to be implemented, as already indicated in Section 2. The stability of the transconductor is easily ensured with the value of the load capacitor selected, as the phase margin ranged between 83.5° and 87.6° when $V_{TUN}$ was swept in the range [200 mV, 600 mV]. The transient behavior to a square wave of the $G_m$ cell connected in unity-gain non-inverting configuration allowed for confirming its stability.

**Table 2.** Aspect ratios (μm/μm) for the main transistors of the transconductor in Figure 7.

| Device | W/L | Device | W/L |
| --- | --- | --- | --- |
| MD1, MD2 | 20/1 | M1, M2, M3, M4 | 1/1 |
| MF1, MF2 | 1/1 | M1C, M2C | 30/0.5 |
| MS1, MS2 | 4/1 | M3C, M4C | 10/0.5 |
| MG1, MG2 | 0.24/0.34 | MR1, MR2 | 1/0.5 |

The robustness of the proposed transconductor has been checked by considering in the simulations mismatches as well as process, voltage, and temperature (PVT) variations. In particular, a 1000-run Monte Carlo analysis with process and mismatch variations in a 3-$\sigma$ range has been carried out. Under these stringent mismatch conditions, the values of the open-loop voltage gain, unity-gain frequency, and phase margin were found to be $45.0 \pm 12.0$ dB, $131.9 \pm 17.9$ kHz, and $83.7 \pm 25.2°$. In addition, the closed-loop BW of the transconductor was $110.0 \pm 24.1$ kHz. In all of these results, the data are represented as the mean value plus/minus the standard deviation. Corner analyses were also run in order to determine the impact of PVT variations on the performance of the transconductor. For the active devices' typical mean (*tt*), fast-fast (*ff*), slow-slow (*ss*) fast-slow (*fs*), and slow-fast (*sf*) conditions were considered, whereas the values of the passive components were varied between the minimum and maximum ranges indicated by the foundry. Additionally, the supply voltage was varied ±10% and the temperature, with nominal value equal to 27 °C, was moved in the range between −20 °C and 80 °C. Considering a total of 45 corners, the open-loop gain, unity-gain frequency, and phase margin varied in the ranges [41.8, 55.6] dB, [84.8, 101.1] kHz, and [84.8, 86.4]°, the closed-loop BW being constrained between 61.4 kHz and 125.4 kHz.

The overall performance of the transconductor is summarized in Table 3, where is it also compared to other similar solutions previously reported. The following figure-of-merit (FoM) has been used for a fair comparison of the transconductors:

$$FoM_T = 100 \cdot \frac{BW \cdot C_L}{P} \qquad (17)$$

where BW is the bandwidth of the transconductor connected in non-inverting unity-gain configuration, $C_L$ is the load capacitor, and $P$ the power consumption. As observed in Table 3, the proposed low-voltage linearized transconductor is competitive in terms of the $FoM_T$, whereas it presents a high open-loop gain at low frequency and provides the largest BW in the comparative.

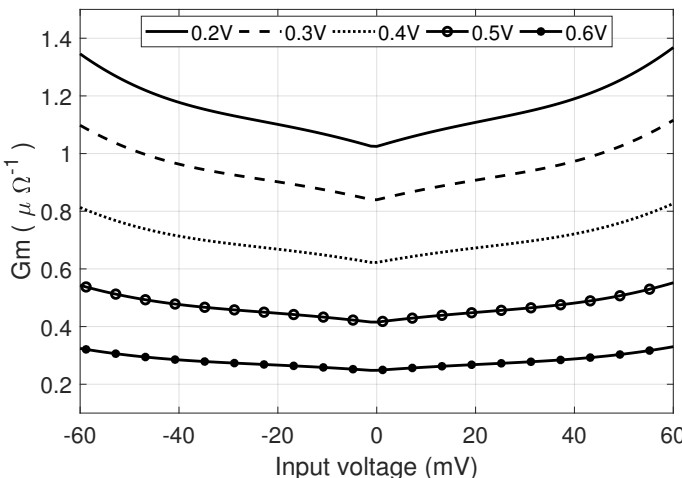

**Figure 11.** Effective transconductance of the linearized transconductor vs. $v_{I,DM}$.

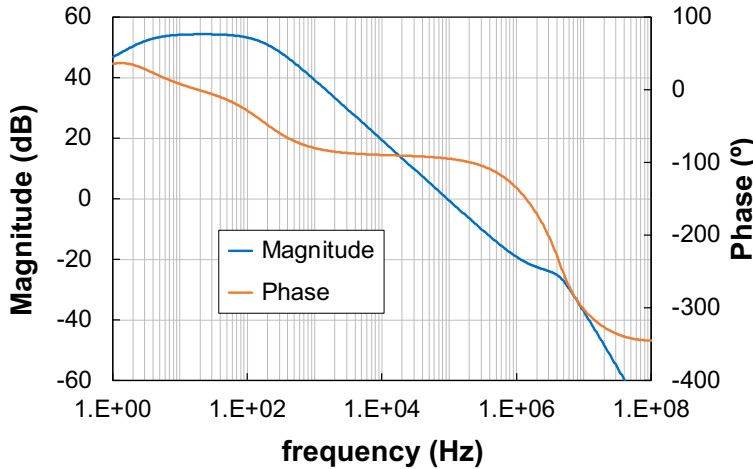

**Figure 12.** Frequency response of the transconductor (left axis: magnitude, right axis: phase).

**Table 3.** Simulated performance of the linearized transconductor and comparison with other similar solutions previously reported.

| Parameter | [17] ALOG'12 | [18] ALOG'14 | [22] Access'21 | [24] TCAS-II'22 | This Work |
|---|---|---|---|---|---|
| Technology (μm) | 0.35 | 0.13 | 0.18 | 0.13 | 0.18 |
| Results | Measured | Measured | Simulated | Measured | Simulated |
| $V_{DD}$ (V) | 0.8 | 0.25 | 0.5 | 0.3 | 0.6 |
| Power (nW) | 40 | 10 | 0.278–535 | 708 | 361.2 |
| $G_m$ (nA/V) | 66 | 22 | 0.34–383 | 4070 | 248.3–1024.9 |
| Open-loop gain (dB) | 61 | NA | 31.2 | 15 | 54.2 |
| BW (kHz) | 0.195 | NA | $2.67 \times 10^{-3}$ | 6 | 99.5 |
| SR$^+$/SR$^-$ (V/ms) | 0.12 | 94600 | NA | NA | 3.15/1.56 |
| THD (dB) | −48.2 @ 600 mV$_{pp}$ | −45.5 @ 100 mV$_{pp}$ | −46.0 @ 480 mV$_{pp}$ | −54.4 @ 100 mV$_{pp}$ | −52.6 @ 200 mV$_{pp}$ |
| $FoM_T$ (kHz·pF/nW) | 12.2 | NA | 19.2-11.5 | 84.7 | 27.5 |

The BPF was implemented by using four transconductors exactly equal excluding the linearization active resistor. Indeed, blocks $G_{m1}$ and $G_{m4}$ have a nominal transconductance nominally equal to $G_m$ and, thus, the sizes of devices MR1 and MR2 correspond to those indicated in Table 2, that is, 1/0.5 μm/μm. Nevertheless, as circuit sections $G_{m2}$ and $G_{m3}$ were sized with a transconductance equal to $4G_m$, transistors MR1 and MR2 in these cases were provided with aspect ratios equal to 3.8/0.5 μm/μm. The biasing current for all the transconductors was set again equal to 100 nA, leading to a total DC power consumption of 2.74 μA. The capacitors in the BPF were implemented as metal–insulator–metal devices, with equal values $C_1 = C_2 = 25$ pF. With these transconductance and capacitor ratios, the quality factor of the BPF was nominally set equal to 4. The reason for selecting relatively high capacitor values is to separate the filter center frequency from the secondary poles of the transconductors, thus avoiding as much as possible any overdamping in the frequency response.

The magnitude response of the BPF over the frequency is depicted in Figure 13 for different values of the tuning variable $V_{TUN}$. As observed, the filter center frequency ranges between 6.5 kHz and 37.5 kHz, which demonstrates that the tuning mechanism results are suitable to avoid the parameter variations due to the fabrication process with a very economical implementation. When $V_{TUN} = V_{BULK} = 400$ mV, the center frequency is equal to 19.1 kHz. The gain of the BPF at the center frequency, nominally set equal to 0 dB as already indicated in (16a), increases slightly as the value of $V_{TUN}$ is decreased, due to the slight overdamping caused by the approaching of $f_0$ to the position of the secondary poles in a system with a relatively high quality factor. The noise of the BPF has been integrated in the −3-dB band for the same tuning conditions previously indicated, obtaining a value of 190.5 μV$_{rms}$. Furthermore, the −40-dB THD criterion has been used to determine the maximum input signal amplitude that can be processed with a given linearity, obtaining a maximum amplitude of 55 mV. At this point, it is interesting to mention that the large value of the time constant associated with capacitor $C_G$ and pseudo-resistor MG in the bootstrapping network leads to a transient response in the BPF output signal of around 1 s before the steady-state regime is achieved. Additionally, the compression curve of the BPF output signal and the third-order intermodulation distortion are represented in Figures 14 and 15, respectively. The IMD3 has been obtained by applying two input tones separated ±100 Hz with respect to the BPF center frequency. In addition, from Figure 14, the input-referred 1-dB compression point has been determined to be −19.13 dBm.

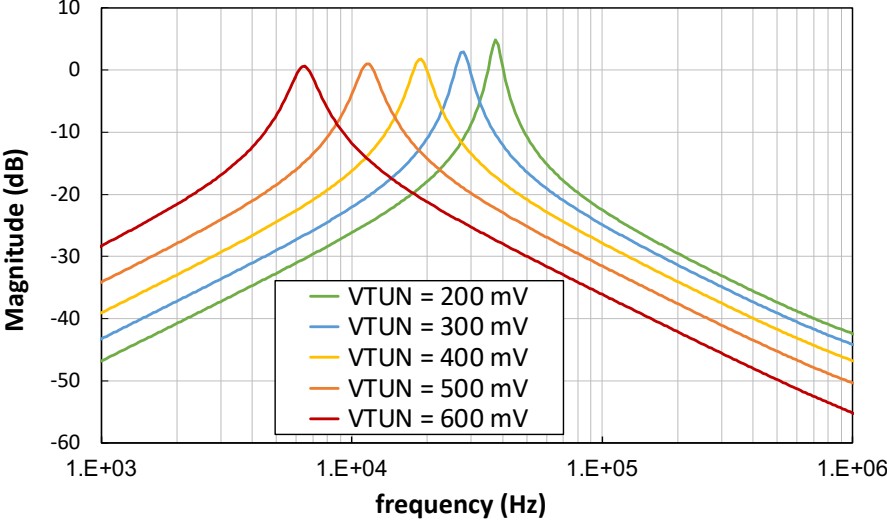

**Figure 13.** Magnitude response vs. frequency of the BPF for different values of $V_{TUN}$.

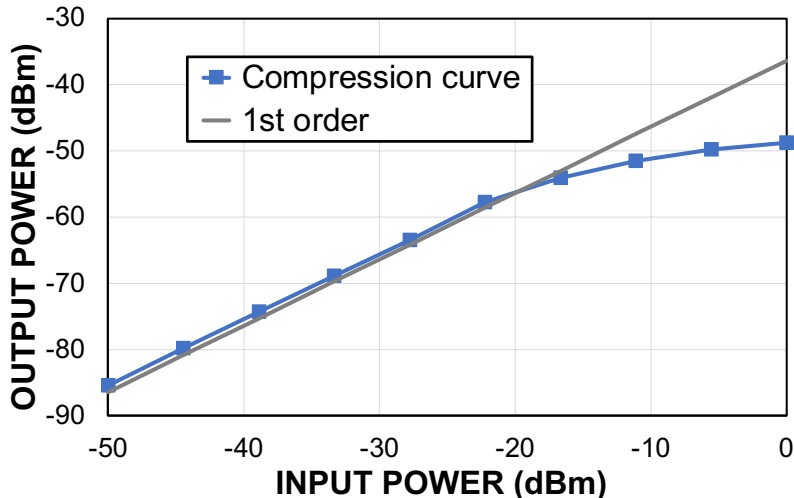

**Figure 14.** Compression curve of the BPF.

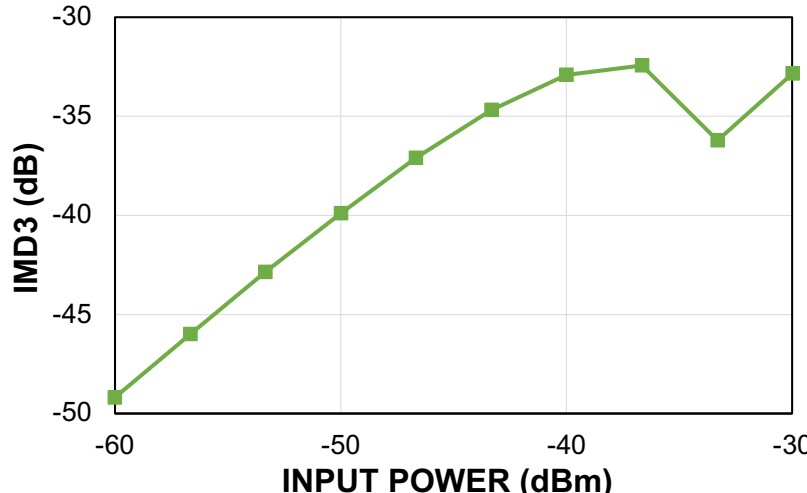

**Figure 15.** IMD3 vs the input signal.

The impact of mismatches and PVT variations on the response of the proposed BPF has been estimated by means of Monte Carlo and corner analyses in the same conditions as described in the case of the linearized transconductor. Regarding Monte Carlo simulations, the center frequency demonstrated itself to be very stable, with a value of $19.4 \pm 1.3$ kHz, showing worst-case responses equal to 16.1 kHz and 20.7 kHz in the corners.

The performance of the proposed BPF is reported in Table 4, where it is compared to other similar solutions previously reported. In order to establish an objective comparison between the different BPF structures, the following FoM has been used [7]

$$FoM_{BPF} = \frac{P \cdot V_{DD}}{n \cdot f_0 \cdot DR} \tag{18}$$

where $P$ is the power consumption, $V_{DD}$ the supply voltage, $n$ the filter order, $f_0$ the center frequency, and DR the dynamic range. It is worth pointing out that the DR has been calculated as the ratio of the input signal leading to a THD of $-40$ dB and the in-band input-referred integrated noise. As observed, the proposed approach features a reduced power consumption in a low supply voltage, which results in being very suitable for bioimpedance-based IoT applications. In addition, the FoM is competitive as compared to the other solutions, with an acceptable DR taking into account the stringent operating conditions at the used supply voltage.

**Table 4.** Simulated performance of the proposed $G_m$-$C$ filter and comparison with similar BPF solutions.

| Parameter | [7]<br>TBCAS'07 | [9]<br>TCAS-II'12 | [10] *<br>MEJ'15 | [14]<br>ICECS'20 | [15] *<br>ICECS'21 | [23]*<br>Access'21 | This Work * |
|---|---|---|---|---|---|---|---|
| Technology (μm) | 0.35 | 0.35 | 0.05 | 0.13 | 0.18 | 0.18 | 0.18 |
| $V_{DD}$ (V) | 1 | 3.3 | 0.4 | 1.2 | 0.8 | 0.5 | 0.6 |
| Power (μW) | 44.3 | 75.4 | 31.8 | 256.0 | 24.0 | 0.06 | 1.65 |
| Filter order | 6 | 2 | 2 | 8 | 2 | 3 | 2 |
| $f_0$ (kHz) | 0.67 | 20 | 10 | 100 | 72.7 | 0.25 | 19.1 |
| $f_0^{min} - f_0^{max}$ (Hz) | ∼100–20 k | 20–20 k | 1–30 k | 2–100 k | 72.7 k | 250 | 6.5–37.5 k |
| $Q$ | N.A | 3 | 1 | 4.8/5.2 | 5 | N.A. | 5.9 |
| $v_{IN,max}$ (m$V_{pp}$) | 40 | 245 ‡ | 178 † | 140 ‡ | 800 | N.A. | 110 † |
| In-band noise (μV) | 70.8 | 58.7 | 53.0 | 100 | 266.6 | 240.0 | 190.5 |
| DR (dB) | 49.0 | 63.5 | 68.4 | 49.0 | 60.5 | 60.4 | 47.4 |
| $FoM_{BPF} \times 10^{-13}$ (SI) | 3.4 | 979.6 | 93.1 | 64.0 | 21.9 | 0.377 | 5.5 |

* Simulated, † @ −40 dB THD, ‡ @ 1-dB compression point.

## 6. Conclusions

The bootstrapping effect has been applied to a bulk-driven MOS transistor in order to enhance its voltage gain up to a value close to unity. As a result, a voltage follower with improved noise and linearity responses and able to operate in extremely low voltage conditions can be obtained. This voltage buffer has been used, along with a low-voltage pseudo-resistor, to implement a linearized transconductor, which is the basic building block of a second-order $G_m$-$C$ BPF aimed at multi-frequency bioimpedance analysis. These circuits have been designed in a 180 nm CMOS process to operate with a supply voltage as low as 0.6 V. The performance of the filter is compatible with the requirements of IoT applications, especially in terms of power consumption, and is comparable to other state-of-the-art solutions previously reported.

**Author Contributions:** Conceptualization, J.M.C. and C.A.d.l.C.-B.; methodology, J.M.C. and C.A.d.l.C.-B.; software, J.M.C. and C.A.d.l.C.-B.; formal analysis, J.M.C. and C.A.d.l.C.-B.; investigation, J.M.C. and C.A.d.l.C.-B.; resources, J.M.C. and C.A.d.l.C.-B.; data curation, J.M.C. and C.A.d.l.C.-B.; writing—original draft preparation, J.M.C. and C.A.d.l.C.-B.; writing—review and editing, J.M.C. and C.A.d.l.C.-B.; visualization, J.M.C. and C.A.d.l.C.-B.; supervision, J.M.C. and C.A.d.l.C.-B.; project administration, J.M.C. and C.A.d.l.C.-B.; funding acquisition, J.M.C. All authors have read and agreed to the published version of the manuscript.

**Funding:** Work funded by projects RTI2018-095994-B-I00, from MCIN/AEI/10.13039/501100011033, and IB18079, from *Junta de Extremadura* R&D Plan, and by Fondo Europeo de Desarrollo Regional (FEDER) Una manera de hacer Europa.

**Data Availability Statement:** Not applicable.

**Conflicts of Interest:** The authors declare no conflict of interests.

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
