# Peer review of "0.6-V 1.65-μW Second-Order Gm-C Bandpass Filter for Multi-Frequency Bioimpedance Analysis Based on a Bootstrapped Bulk-Driven Voltage Buffer"

_jlpea, doi:10.3390/jlpea12040062_

Round 1

Reviewer 1 Report

The manuscript presents Second-Order Gm-C Band pass Filter for Multi-Frequency Bio impedance Analysis Based on a Bootstrapped Bulk-Driven Voltage Buffer.

The paper in general is interesting and well written. However, I have the following comments:

-          Since the paper show a BPF please show a figure of intermodulation distortion IMD3 versus input voltage.

-          Since the paper deals with the design of the transconductor please consider adding a  comparison table with recently published transconductor (mentioned below).

-          Fig. 13 what is the reason of increasing the gain with decreasing the VTUN?

-          Since the spectrum of the biomedical signals lie between sub hertz up to 10 kHz, please add a figure showing the center frequency of the BPF versus VTUN.

-          How the DR is calculated?

-           Please include the PVT and MC analysis for the transconductor

-          References are old and should be updated including the following related references :

1.         “A 0.3-V high linear rail-to-rail bulk-driven OTA in 0.13 µm CMOS,” IEEE Trans.Circuits Syst.—II Express Briefs 2022, 69, 2046–2050. DOI: 10.1109/TCSII.2022.3144095

2.         “A 0.5-V multiple-input bulk-driven OTA in 0.18-μm CMOS,” IEEE Transactions on Very Large Scale Integration (VLSI) Systems, 2022, DOI: 10.1109/TVLSI.2022.3203148.

3.          “0.3-V Nanopower Biopotential Low-Pass Filter” IEEE Access, vol. 8, pp. 119586-119593, 2020, DOI: 10.1109/ACCESS.2020.3005715

4.         “0.3-V Bulk-Driven Nanopower OTA-C Integrator in 0.18 μm CMOS”, Circuits, Systems, and Signal Processing 2019, 38, pp. 1333–1341, DOI: 10.1007/s00034-018-0901-x

Author Response

Please, see attached file. Thank you!

Reviewer 2 Report

The paper describes a bootsrapping technique used to increase the intrinsic voltage gain of a bulk-driven MOS transistor. The proposed circuit incorporates a capacitor and a cutoff transistor to be connected to the gate terminal of a bulk-driven MOS device, thus achieving a quasi-floating-gate structure. As a result, the contribution of the gate transconductance is cancelled out and the voltage gain of the device is correspondingly increased. The technique allows implementing a voltage follower with a voltage gain much closer to unity as compared to the conventional bulk-driven case. This voltage buffer, along with a pseudo-resistor, is used to design a linearized transconductor. The proposed transconductance cell includes an economic continuous tuning mechanism that permits programming the effective transconductance in a range sufficiently wide to counteract the typical variations that process parameters suffer during fabrication. The transconductor has been used to implement a second-order Gm-C bandpass filter with a relatively high selectivity factor, suited for multi-frequency bioimpedance analysis in a very low-voltage environment. All the circuits have been designed in 180 nm CMOS technology to operate with a 0.6-V single-supply voltage. Simulated results show that the proposed technique allows increasing the linearity and reducing the input-referred noise of the bootstrapped bulk-driven MOS transistor, which results in an improvement of the overall performance of the transconductor. The center frequency of the bandpass filter designed can be programmed in the frequency range from 6.5 kHz to 37.5 kHz with a power consumption ranging between 1.34 µW and 2.19 µW. The circuit presents an in-band integrated noise of 190.5 µVrms and is able to process signals of 110 mVpp with a THD below −40 dB, thus leading to a dynamic 19 range of 47.4 dB.

Generally speaking the paper is interesting and well-structured. It is completed of all the minimum information, like an exhaustive introduction, method and circuital explanation, design flow and measured results on a prototype. It is well-written and easy to follow.

I have only two suggestions:

1- I would suggest the following three recent works which aim with the topic of low-voltage amplifiers and second-order band-pass filter.

a) Active load with cross‐coupled bulk for high‐gain high‐CMRR nanometer CMOS differential stages

b) Area-Efficient Low-Power Bandpass Gm-C Filter for Epileptic Seizure Detection in 130nm CMOS

c) High-frequency low-current second-order bandpass active filter topology and its design in 28-nm FD-SOI CMOS

2 - It is surely interesting if the transient response and the 1-dB compression point curve and third-order intercept point curves could be provided.

Author Response

Please, see attached file. Thank you!

Round 2

Reviewer 1 Report

The authors have addressed all my comments. I have the following minor comment:

For practical applications the low temperature corner could be much less than 0C, let’s say -20C or even -40C, why the 0C is used? Is there any reason? If so please provide on the text otherwise please use less temperature than 0C e.g. -20C.

Author Response

Please, see attached file. Thank you!
